# Positive allosteric modulation of the α7 nicotinic acetylcholine receptor as a treatment for cognitive deficits after traumatic brain injury

**David J. Titus**[1], **Timothy Johnstone**[2], **Nathan H. Johnson**[1], **Sidney H. London**[1], **Meghana Chapalamadugu**[1], **Derk Hogenkamp**[2], **Kelvin W. Gee**[2], **Coleen M. Atkins**[1] *

**1** Department of Neurological Surgery, The Miami Project to Cure Paralysis, University of Miami Miller School of Medicine, Miami, Florida, United States of America, **2** Department of Pharmacology, School of Medicine, University of California Irvine, Irvine, California, United States of America

\* catkins@miami.edu

**Data Availability Statement:** The data underlying the results presented in the study are available

## Abstract

Cognitive impairments are a common consequence of traumatic brain injury (TBI). The hippocampus is a subcortical structure that plays a key role in the formation of declarative memories and is highly vulnerable to TBI. The α7 nicotinic acetylcholine receptor (nAChR) is highly expressed in the hippocampus and reduced expression and function of this receptor are linked with cognitive impairments in Alzheimer's disease and schizophrenia. Positive allosteric modulation of α7 nAChRs with AVL-3288 enhances receptor currents and improves cognitive functioning in naïve animals and healthy human subjects. Therefore, we hypothesized that targeting the α7 nAChR with the positive allosteric modulator AVL-3288 would enhance cognitive functioning in the chronic recovery period of TBI. To test this hypothesis, adult male Sprague Dawley rats received moderate parasagittal fluid-percussion brain injury or sham surgery. At 3 months after recovery, animals were treated with vehicle or AVL-3288 at 30 min prior to cue and contextual fear conditioning and the water maze task. Treatment of TBI animals with AVL-3288 rescued learning and memory deficits in water maze retention and working memory. AVL-3288 treatment also improved cue and contextual fear memory when tested at 24 hr and 1 month after training, when TBI animals were treated acutely just during fear conditioning at 3 months post-TBI. Hippocampal atrophy but not cortical atrophy was reduced with AVL-3288 treatment in the chronic recovery phase of TBI. AVL-3288 application to acute hippocampal slices from animals at 3 months after TBI rescued basal synaptic transmission deficits and long-term potentiation (LTP) in area CA1. Our results demonstrate that AVL-3288 improves hippocampal synaptic plasticity, and learning and memory performance after TBI in the chronic recovery period. Enhancing cholinergic transmission through positive allosteric modulation of the α7 nAChR may be a novel therapeutic to improve cognition after TBI.

## Introduction

There are an estimated 3.17 million people in the US coping with long-term disabilities due to TBI, resulting in an economic burden exceeding $56 billion annually [1, 2]. Nearly 80% of

from the Dryad Digital Repository: https://doi.org/
10.5061/dryad.803qk45.

**Funding:** This work was supported by a grant
awarded to C.M.A. by the NIH/NINDS NS069721
and the Miami Project to Cure Paralysis.

people who have sustained a TBI report learning and memory impairments in the months to
years after the initial brain trauma [3]. The hippocampus, an area of the brain important for
declarative memory formation, is highly vulnerable and exhibits a significant degree of pathology after TBI [4, 5]. Although there is some understanding of the changes within the hippocampus as a consequence of TBI, no FDA-approved pharmacological therapies are available to
ameliorate cognitive deficits in the chronic recovery phase of TBI [6]. The development of a
robust, efficacious therapeutic to target these alterations in the hippocampus and improve
learning and memory is an area that needs investigation.

Neuronal circuits involved in cognition, attention and concentration contain nicotinic cholinergic synapses [7, 8]. These circuits are altered with age and are associated with cognitive
decline and Alzheimer's disease [9, 10]. The most abundant nAChRs in the brain are the α7 and
α4β2 subunit-containing receptors [11, 12]. In particular, the α7 nAChR is found both pre- and
post-synaptically on excitatory neurons in the hippocampus as well as on interneurons [13, 14].
The α7 nAChR modulates fast synaptic transmission, neurotransmitter release and synaptic
plasticity due to its ability to gate calcium flux [15–17]. Activation of α7 nAChRs converts
short-term potentiation into LTP, which is particularly relevant for experimental TBI, where
short-term potentiation typically remains intact, but maintenance of LTP is impaired [18–21].

Numerous preclinical studies have established that in the days to weeks after TBI, there is a
decrease in cholinergic signaling [22–24]. There is reduced high-affinity choline uptake [25],
decreased choline acetyltransferase activity [26], acute reductions in vesicular acetylcholine
transporter [27] and transient depression of cholinesterase activity in the hippocampus [28].
At the receptor level, there is a loss of up to 50% of α7 nAChRs after controlled cortical impact
[29, 30]. This decrease is accompanied by a bilateral loss of cholinergic neurons and their axonal projections [31]. Decreased cholinergic signaling has been reported in human TBI studies
as well, lasting for at least one year after trauma [32, 33]. Although cholinergic signaling is
decreased chronically after TBI, it is not completely absent. This suggests that therapeutics that
enhance the remaining endogenous cholinergic activity may be efficacious.

It is well established that agonists of cholinergic receptors rescue memory impairments in preclinical models of TBI [34–36]; however, results with cholinesterase inhibitors have had mixed success [37–46]. Cholinesterase inhibitors act by blocking the hydrolysis of acetylcholine which in
turn increases acetylcholine levels. Chronic administration of either full or partial agonists, or cholinesterase inhibitors may be limited by the natural adaptation (e.g., receptor desensitization) of the
brain to increased levels of agonist. A method to preserve spatial and temporal integrity of cholinergic signaling is to utilize allosteric modulators. Positive allosteric modulators of the α7 nAChR
bind the receptor, but only enhance receptor current when the receptor is bound to its agonist.
Positive allosteric modulators for α7 nAChRs are divided into two types, depending on whether
they have no effect on receptor desensitization kinetics (type I) or slow desensitization (type II).
This differentiation of mechanism is important since the α7 nAChR gates calcium flux and naturally desensitizes rapidly. In this study, we investigated the actions of a type I positive allosteric
modulator of the α7 nAChR, AVL-3288, on outcome in the chronic recovery phase of TBI [47].

## Materials and methods

### Animals

All animal procedures were in compliance with the NIH Guide for the Care and Use of Laboratory Animals and approved by the University of Miami Animal Care and Use Committee. Adult
male Sprague-Dawley rats ($n = 94$ total, 2–3 months old, Charles Rivers Laboratories) were
maintained on a 12/12 hr light/dark cycle and had free access to food and water. To determine
the minimum number of animals needed for these studies, a power analysis was prospectively

performed to detect a 20% difference in water maze probe trial performance between groups at 80% power with a significance level of 0.05 [19]. A sample size of 11 animals/group was obtained. The effect size for our observed differences in the water maze probe trial was 1.28, recent recall of contextual fear conditioning was 1.39, remote recall of contextual fear conditioning was 0.92 and LTP expression at 30–60 post-tetanization was 1.27. These large effect sizes indicate that our sample size was adequate for the main objective of this study, to determine if AVL-3288 improved hippocampal learning and memory and LTP in TBI animals.

## Fluid-percussion injury surgery

Animals were randomly assigned prior to the experiment to receive either moderate parasagittal fluid-percussion injury (FPI) or sham surgery, and either treatment with vehicle or AVL-3288. Under deep anesthesia (3% isoflurane, 70% $N_2O$, and 30% $O_2$, maintained at 1–2% isoflurane, 70% $N_2O$, and 30% $O_2$), a 4.8 mm craniotomy was made at 3.8 mm posterior to bregma, 2.5 mm right from midline and a beveled 18 gauge syringe hub was secured to the craniotomy site. At 12–16 hr after the craniotomy, animals were re-anesthetized, intubated and mechanically ventilated (Stoelting) with 1–2% isoflurane, 70% $N_2O$, and 30% $O_2$. All animals received a catheter in the tail artery for physiological monitoring. Rocuronium (10 mg/kg) was administered through the tail artery to facilitate ventilation, and penicillin G potassium (20,000 IU/kg, intramuscular) was given during the surgery. A moderate (1.9±0.1 atm) fluid-percussion pulse (14–16 msec duration) was delivered to the right parietal cortex. Sham-operated rats received all surgical manipulations except for the fluid pulse. Anesthesia duration during the surgeries was monitored to ensure that the cumulative duration of anesthetic was consistent across treatment groups. Each animal received two anesthesia rounds. The anesthetic duration for each surgery was less than 1 hr, and animals were not behaviorally tested until 3 months after the final surgery. Rectal and temporalis muscle thermistors were used to maintain body and head temperatures between 36.6–37.2˚C. Mean arterial blood pressure (MAPB) was monitored continuously during the surgery with a transducer connected to the tail artery catheter and LabChart 7 software (ADInstruments). Blood gases and blood pH were measured from blood harvested from the tail artery with a blood gas analyzer (ABL800 Flex, Radiometer America). Atmospheres of pressure were measured using a transducer connected to the fluid percussion injury shaft and LabChart 7 software. Blood gases ($pO_2$ and $pCO_2$), blood pH and MABP were maintained within normal physiological ranges (Table 1). No significant differences were observed in these physiological measures (body weight, MABP, blood $pO_2$, blood $pCO_2$, blood pH, body temperature, head temperature) between Sham+Vehicle, Sham+AVL-3288, TBI+Vehicle or TBI+AVL-3288 treated animals. Buprenorphine (0.01 mg/kg, subcutaneously) was administered at the completion of the surgery. Exclusion criteria were: mortality, >15% loss of body weight, non-resolving infection at a surgical site, inability to feed or drink, motor paralysis, listlessness, self-mutilation, excessive grooming leading to loss of dermal layers, excessive spontaneous vocalization when touched, or poor grooming habits. Animals were monitored daily after surgery for the first 2 days, then evaluated and weighed every 2 weeks until perfusion or decapitation. Attrition for sham surgery was 0% and for TBI surgery was 1% (1 animal, which died at the time of surgery due to lung edema). Investigators were blind to the animal surgery allocation and drug treatment for all behavior, electrophysiology and histology analyses.

## Drug administration

AVL-3288 was synthesized as described [48]. AVL-3288 (0.3 mg/kg) or vehicle (2% DMSO, 8% Solutol and 90% saline, 1 ml/kg) were administered intraperitoneally at 30 min prior to:

**Table 1. Physiological data.**

| Parameter | Treatment | At surgery | At perfusion | Parameter | Treatment | 15 min prior to FPI | 15 min post-FPI |
|---|---|---|---|---|---|---|---|
| **Weight** (gm) | Sham+Vehicle | 359.3±10.8 | 666.0±22.6 *** | **MABP** (mmHg) | Sham+Vehicle | 121.3±3.6 | 121.6±2.2 |
| | Sham+AVL-3288 | 363.4±16.5 | 643.5±19.0 *** | | Sham+AVL-3288 | 122.8±4.1 | 111.2±4.2 |
| | TBI+Vehicle | 349.5±10.8 | 623.5±20.9 *** | | TBI+Vehicle | 115.7±3.5 | 113.0±2.2 |
| | TBI+AVL-3288 | 360.3±13.0 | 614.8±22.0 *** | | TBI+AVL-3288 | 116.8±4.5 | 110.8±3.8 |
| **ATM** | Sham+Vehicle | N/A | | **Blood $pO_2$** (mmHg) | Sham+Vehicle | 153.4±6.6 | 150.7±5.0 |
| | Sham+AVL-3288 | N/A | | | Sham+AVL-3288 | 158.0±9.9 | 147.8±6.9 |
| | TBI+Vehicle | 1.9±0.0 | | | TBI+Vehicle | 153.9±6.8 | 136.3±5.0 * |
| | TBI+AVL-3288 | 1.9±0.0 | | | TBI+AVL-3288 | 149.8±6.5 | 134.5±5.6 ** |

| Parameter | Treatment | 15 min prior to FPI | 15 min post-FPI | Parameter | Treatment | 15 min prior to FPI | 15 min post-FPI |
|---|---|---|---|---|---|---|---|
| **Blood $pCO_2$** (mmHg) | Sham+Vehicle | 41.9±1.1 | 40.2±0.9 * | **Head temperature** (ºC) | Sham+Vehicle | 36.6±0.0 | 36.8±0.0 ** |
| | Sham+AVL-3288 | 39.8±0.9 | 38.1±0.7 * | | Sham+AVL-3288 | 36.7±0.0 | 36.8±0.0 ** |
| | TBI+Vehicle | 38.5±0.8 | 36.8±0.5 * | | TBI+Vehicle | 36.7±0.0 | 36.8±0.0 |
| | TBI+AVL-3288 | 39.0±0.7 | 37.2±0.5 * | | TBI+AVL-3288 | 36.6±0.1 | 36.7±0.0 |
| **Blood pH** | Sham+Vehicle | 7.4±0.0 | 7.5±0.0 ** | **Body temperature** (ºC) | Sham+Vehicle | 36.8±0.0 | 36.7±0.0 |
| | Sham+AVL-3288 | 7.5±0.0 | 7.5±0.0 | | Sham+AVL-3288 | 36.7±0.0 | 36.8±0.0 |
| | TBI+Vehicle | 7.4±0.0 | 7.5±0.0 | | TBI+Vehicle | 36.8±0.1 | 36.8±0.0 |
| | TBI+AVL-3288 | 7.5±0.0 | 7.5±0.0 | | TBI+AVL-3288 | 36.8±0.1 | 36.8±0.0 |

ATM: atmospheres of pressure, MABP: mean arterial blood pressure, N/A: not applicable, $pO_2$: partial arterial oxygen pressure, $pCO_2$: partial arterial carbon dioxide pressure.

*$p < 0.05$

**$p < 0.01$

***$p < 0.001$ vs. at surgery or 15 min prior to FPI, repeated measures two-way ANOVA with Tukey's HSD post-hoc test. Mean±SEM, $n = 10$ Sham+Vehicle, n = 8 Sham+AVL-3288, $n = 11$ TBI+Vehicle, $n = 11$ TBI+AVL-3288.

training on cue and contextual fear conditioning, water maze training days 1–4, both testing days for working memory, and shock threshold testing. Each animal received a total of 8 vehicle or AVL-3288 treatments. Retention at 24 hr after cue and contextual fear conditioning and the probe trial of the water maze were assessed without drug treatment to evaluate whether AVL-3288 had effects on memory retention beyond the acute treatment period. Hippocampal slices were treated with AVL-3288 (1 μM) or vehicle (0.01% DMSO) for 10 min prior to high-frequency tetanization and for 30 min after LTP induction.

## Pharmacokinetic analysis

Naïve, non-injured animals received AVL-3288 (in 2% DMSO, 8% Solutol, 90% saline, 0.3 mg/kg, intraperitoneally) and then were deeply anesthetized (3% isoflurane, 70% $N_2O$, and 30% $O_2$) from 15 min to 24 hr after treatment. Whole brain tissue and trunk blood were collected. Plasma was prepared with 0.5M $K^+$-EDTA, and centrifuged at 3000 x g for 10 min at 4˚C. The brains were extracted with ethyl acetate. AVL-3288 levels were measured by LC-MS/MS. The chromatographic separation was performed using a Luna, C8, 2.0 x 150 mm 5 μm column (Phenomenex) with 75:25 acetonitrile:0.1% formic acid in water mobile phase with a flow rate of 0.3 ml/min at 40˚C with an Agilent 1100/1200 HPLC and an Agilent 6410B QQQ Mass Spectrometer (Agilent Technologies). The mass spectrometer parameters were: FV 100, CE 8, with positive ion monitoring. The pharmacokinetic data were analyzed by standard methods using Agilent Mass Hunter Software Version 4.0, SAS JMP Version 7 and Microsoft Excel.

## Fear conditioning

At 3 months post-surgery, animals were trained and tested for cue and contextual fear conditioning [19]. Animals were first habituated for 10 min to a cage equipped with a shock grid floor (Coulbourn Instruments). At 24 hr after habituation, animals were trained by being placed in the apparatus for 120 s, and then a 30 s tone (75 dB, 2.8 kHz) was delivered that co-terminated with a 1 mA, 1 s foot shock. Animals remained in the apparatus for 60 s after the foot shock. At 24 hr and 1 month after training, animals were placed in the apparatus and freezing was measured for 5 min to assess contextual fear conditioning. Cue fear conditioning was evaluated 1 hr after assessment of contextual fear by placing the animals in an altered chamber. The tone (75 DB, 2.8 kHz) was played for 60 s and freezing was measured. Freezing behavior was quantified by video analysis (FreezeFrame 3.32, Coulbourn Instruments). Shock threshold was assessed 1 day after final cue and contextual fear conditioning testing (4 months post-surgery). Animals received a 1 s foot shock every 30 s in increments of 0.02 mA beginning at 0.1 mA. Minimum shock intensity to elicit a flinch, jump, or vocalization was measured.

## Water maze

At 1 week after cue and contextual fear conditioning (13 weeks post-surgery), animals were next trained in the water maze for 4 trials/day (60 s maximum trial duration) over 4 days with an inter-trial interval of 4–6 min [19]. Path length to reach the platform was measured by video analysis (EthoVision XT 10, Noldus Information Technology). At 24 hr after the final acquisition day, a probe trial (60 s duration) was given with the platform removed. The amount of time spent in each quadrant of the pool was measured.

At 1 week after water maze training and testing (14 weeks post-surgery), working memory was assessed in the water maze [19]. Animals received 2 days of testing with the platform remaining invariant between each pair of trials (5 s delay between trials, 60 s maximum trial duration). Each day consisted of 4 paired trials (4–6 min between each trial pair). Path length to reach the platform was measured. Data from day 2 were analyzed.

## Histology

At the completion of behavioral testing (4–5 months post-surgery), animals were deeply anesthetized (3% isoflurane, 70% $N_2O$, 30% $O_2$) and transcardially perfused with saline and 4% paraformaldehyde in 0.1M phosphate buffer, pH 7.4. Brains were paraffin-embedded and sectioned (10 μm thick, 150 μm apart) and stained with hematoxylin and eosin (H&E) plus Luxol fast blue as previously described [19]. Sections were imaged at 7200 dpi (3.5 μm/pixel) with a Quick Scan PathScan Enabler IV 3.60.0.12 (Meyer Instruments). The cortex and hippocampus were contoured between bregma levels -3.3 to -6.8 mm using Neurolucida 11.11.2 (MicroBrightField). Atrophy was quantified by subtracting the contralateral volume from the ipsilateral volume and normalizing to the contralateral volume. Images were obtained on an Olympus BX51TRF microscope (Olympus America) with a 20× objective and stitched with StereoInvestigator 5.65 software (MicroBrightField). Three-dimensional projections were rendered in NeuroLucida 11.11.2. To construct the 3-dimensional projections, the outer shells of the brain contours were aligned using the rhinal fissures and the midline. Cortical contours were aligned using the retrosplenial granular area and hippocampal contours were aligned using the habenular nuclei and dentate gyrus.

## Electrophysiology

At 3 months post-surgery, animals were deeply anesthetized (3% isoflurane, 70% $N_2O$, and 30% $O_2$), decapitated and the hippocampus was quickly isolated from the brain. Transverse

slices (400 μm thick) were prepared as previously described [19]. Recordings were made in aCSF: 125 mM NaCl, 2.5 mM KCl, 1.25 mM $NaH_2PO_4$, 25 mM $NaHCO_3$, 10 mM D-glucose, 2 mM $CaCl_2$, 1 mM $MgCl_2$, saturated with 95% $O_2$/5% $CO_2$ at room temperature. The Schaffer collateral pathway was stimulated with a platinum-iridium cluster stimulating electrode (tip diameter 25 μm, FHC). Field excitatory postsynaptic potentials (fEPSPs) and fiber volleys were recorded from CA1 stratum radiatum with glass micropipettes filled with 2 M NaCl (1–3 MΩ) using a Multiclamp 700B amplifier (Axon Instruments). Signals were low pass filtered at 2 kHz and digitized at 20 kHz with a Digidata 1440A interface and analyzed with pClamp 11 (Axon Instruments). Input-output (I-O) curves were generated by stimulating from 20–240 μA. Paired-pulse facilitation (PPF) was measured with stimulus intervals from 12.5–250 msec, with stimulation intensity set at 40–50% of the maximum fEPSP. Prior to LTP induction, baseline responses were recorded at 40–50% of the maximum fEPSP at 0.033 Hz for at least 20 min. LTP was induced by high-frequency stimulation (HFS) using a single train of 100 pulses delivered at 100 Hz at test stimulation intensity. AVL-3288 (1 μM) or vehicle (0.01% DMSO) were bath-applied in aCSF beginning 10 min prior to HFS and for 30 min after HFS. The tetanization response was analyzed by integrating the entire HFS response and also by integrating the depolarization during last 50 msec of the HFS to determine steady-state depolarization [49]. Synaptic fatigue was calculated by measuring each fEPSP during the HFS and normalizing this to the first fEPSP of the HFS [50].

## Statistical analysis

Data presented are mean ± SEM. Significance was designated at $p < 0.05$. Statistical analyses were performed in GraphPad Prism 7.0 and SigmaPlot 12.0. Two-way ANOVA and repeated measures two-way ANOVA were used to determine group differences with the factors surgery x drug treatment or animal group x time/trial/current intensity. Tukey's HSD post-hoc tests were used when significant interactions were observed. Simple linear regression fits were used to evaluate the fiber volley amplitude and fEPSP slope and the slopes were compared with a two-way ANOVA with the factors surgery x drug treatment.

## Results

### AVL-3288 levels in brain and plasma

AVL-3288 selectively enhances wildtype, human α7 nAChR currents evoked by nicotine in *Xenopus* oocytes at a dose as low as 0.1 μM with an $I_{max}$ of 3 μM [47]. To determine if AVL-3288 could be singly administered systemically and reach levels in the brain that are effective in modulating α7 nAChR currents, levels of AVL-3288 in the brain and plasma were measured after intraperitoneal administration (Fig 1). Naïve, non-injured rats received AVL-3288 (0.3 mg/kg, intraperitoneally) and then plasma and brain tissue were analyzed using LC-MS/MS as previously described [51]. AVL-3288 reached 3 μM in total brain tissue within 30 min of administration and peaked at 4.6 μM by 90 min. Levels returned to baseline by 24 hr after administration. Based on these results and a previous study demonstrating that the minimum effective dose to improve radial arm maze memory in rats was 0.3 mg/kg AVL-3288, we chose to use this dose in our TBI study [47].

### Hippocampal-dependent learning and memory ability is improved with AVL-3228 treatment after TBI

This preclinical model of TBI is characterized by persistent hippocampal-dependent learning and memory deficits in contextual fear conditioning and the water maze task [19, 52]. To

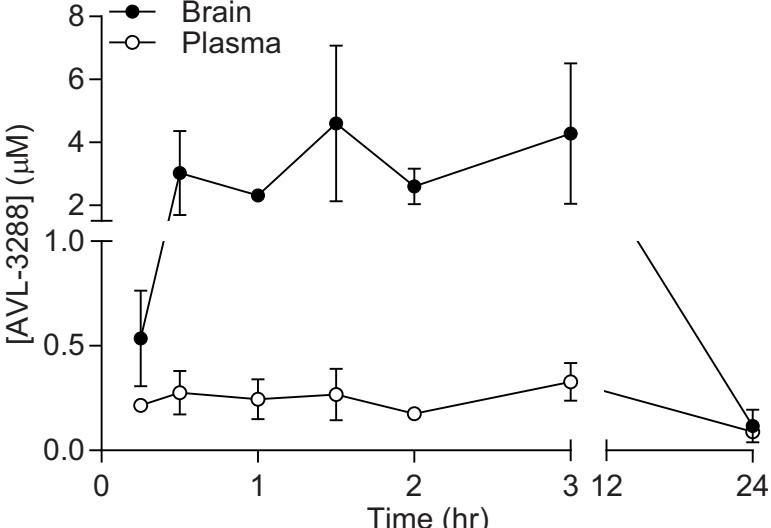

**Fig 1. Pharmacokinetics.** Levels (μM) of AVL-3288 in the brain and plasma. Naïve, non-injured animals were treated with AVL-3288 intraperitoneally (0.3 mg/kg in 2% DMSO, 8% Solutol, 90% saline). AVL-3288 was measured in total brain tissue and plasma by LC-MS/MS. Mean ± SEM, $n$ = 3/group.

determine if AVL-3288 could rescue these cognitive deficits, animals were treated with AVL-3288 or vehicle at 30 min prior to cue and contextual fear conditioning (Fig 2). Since activation of α7 nAChRs converts short-term potentiation in hippocampal slices to LTP, we chose to treat animals during training, but not during testing [18–21]. No significant differences were observed in cue and contextual fear conditioning between sham animals treated with vehicle versus AVL-3288. In TBI animals, AVL-3288 significantly improved both cue and contextual fear conditioning when assessed 24 hr after training for recent recall (surgery x drug treatment interaction: cue $F_{(1,36)}$ = 4.29, $p$ = 0.045; context $F_{(1,36)}$ = 8.32, $p$ = 0.007). The effects on fear memory were persistent as shown by a significant improvement in remote recall assessed 1 month after training (surgery x drug treatment interaction: cue $F_{(1,36)}$ = 6.75, $p$ = 0.013; context $F_{(1,36)}$ = 6.59, $p$ = 0.015).

Next, to determine if AVL-3288 improves other types of hippocampal-dependent learning, animals were tested in the water maze to find a hidden platform using spatial cues (Fig 3A–3E). Animals were pretreated with AVL-3288 at 30 min prior to water maze acquisition for 4 days. AVL-3288 treatment did not significantly reduce the path length to find the platform on the final day of acquisition (day 4) in TBI animals as compared to vehicle-treated TBI animals. Retention at 24 hr after the last acquisition day was assessed without drug treatment to determine if AVL-3288 needed to be present for an effect on memory retention. TBI animals treated with vehicle demonstrated no significant preference for the target quadrant. In contrast, sham animals treated with either vehicle or AVL-3288, and TBI animals treated with AVL-3288 searched the target quadrant significantly more than the other quadrants, indicating that spatial memory was improved with AVL-3288 treatment (surgery x drug treatment interaction for time in target quadrant: $F_{(1,36)}$ = 5.00, $p$ = 0.032).

Assessment of spatial working memory revealed similar improvements with AVL-3288 treatment (Fig 3F). Animals were trained on a delay match-to-sample task in the water maze, with 5 s between a location and match trial and the platform remaining in the same location only during pairs of trials. Sham+Vehicle, Sham+AVL-3288 and TBI+AVL-3288 treated animals had significantly shorter path lengths to find the platform on the match trial as compared

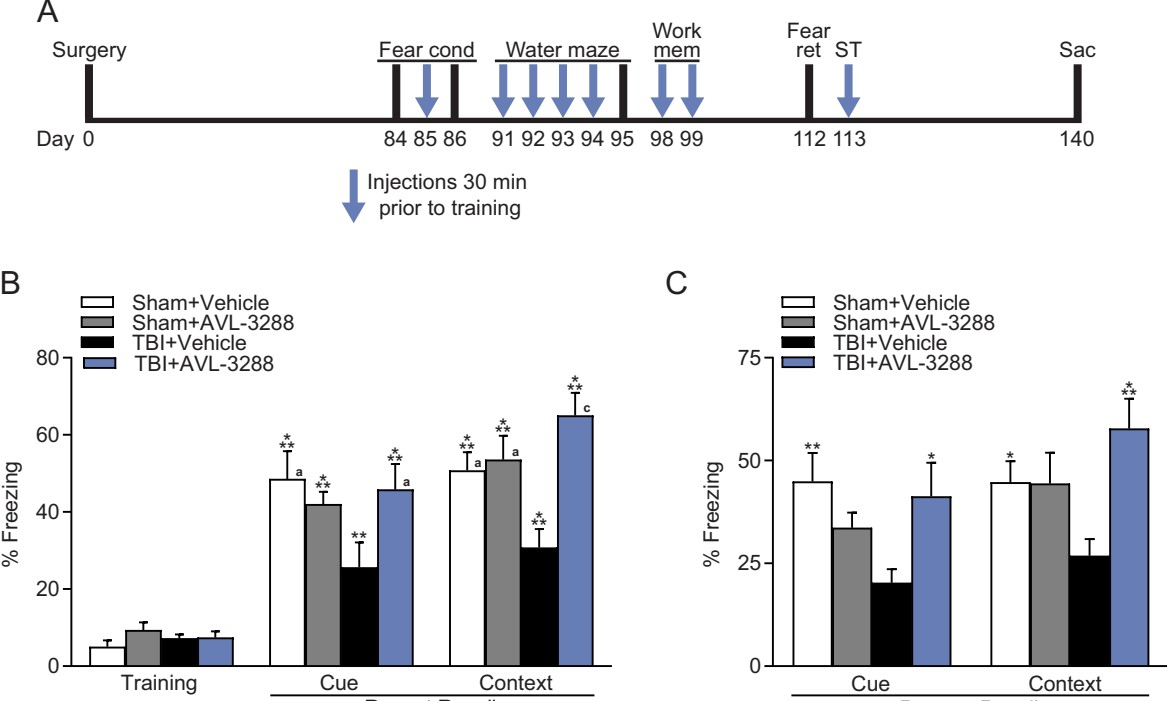

**Fig 2. Fear conditioning.** Cue and contextual fear conditioning were improved with AVL-3288 treatment in TBI animals. A) Time course of AVL-3288 treatment (*arrows*) and behavioral analyses. Animals were serially tested on fear conditioning and then on the water maze tasks. Animals received AVL-3288 or vehicle treatment 30 min prior to training for fear conditioning (Fear cond), on the 4 training days of the water maze, on each day of training for working memory (Work mem), and during shock threshold testing (ST) for a total of 8 treatments over the course of 1 month as depicted by the *arrows*. Animals were tested without AVL-3288 treatment for memory retention of fear conditioning at 24 hr and 1 month post-training (Fear ret) and on the probe trial for the water maze at 24 hr post-training. B) Recent recall of fear conditioning at 24 hr after training. **$p<0.01$, ***$p<0.001$ vs. Training, $^a p<0.05$, $^c p<0.001$ vs. TBI+Vehicle, repeated measures two-way ANOVA with Tukey's HSD post-hoc test. C) Assessment of fear conditioning at one month after training to evaluate remote recall of fear memory. *$p<0.05$, **$p<0.01$, ***$p<0.001$ vs. TBI+Vehicle, two-way ANOVA with Tukey's HSD post-hoc test. Mean ± SEM, $n$ = 10 Sham+Vehicle, $n$ = 8 Sham+AVL-3288, $n$ = 11 TBI+Vehicle, $n$ = 11 TBI+AVL-3288.

to the location trial, indicative of working memory, whereas TBI+Vehicle treated animals had significantly longer paths to find the platform on the match trial (animal group x trial interaction: $F_{(3,32)} = 3.25$, $p = 0.035$).

## Effects on hippocampal atrophy with AVL-3288 treatment after TBI

Previous studies have demonstrated that choline, nicotine and donepezil administration not only improve cognition, but also reduce lesion volume and rescue CA1 neuronal loss after TBI [34, 36, 53, 54]. At the completion of behavioral testing, we evaluated cortical and hippocampal atrophy which is characteristic of this preclinical model of TBI at this recovery time point [55] (Fig 4). Cortical atrophy in the ipsilateral parietal cortex was similar between TBI animals treated with vehicle and AVL-3288. However, hippocampal atrophy on the ipsilateral side was modestly reduced in TBI animals treated with AVL-3288 as compared to TBI animals treated with vehicle (surgery x drug treatment interaction: $F_{(1,36)} = 5.81$, $p = 0.021$).

## Positive allosteric modulation of α7 nAChR enhances basal synaptic transmission and LTP after chronic TBI

To assess electrophysiological changes with AVL-3288 treatment, a separate cohort of animals received sham surgery or moderate FPI and then were allowed to recover without any

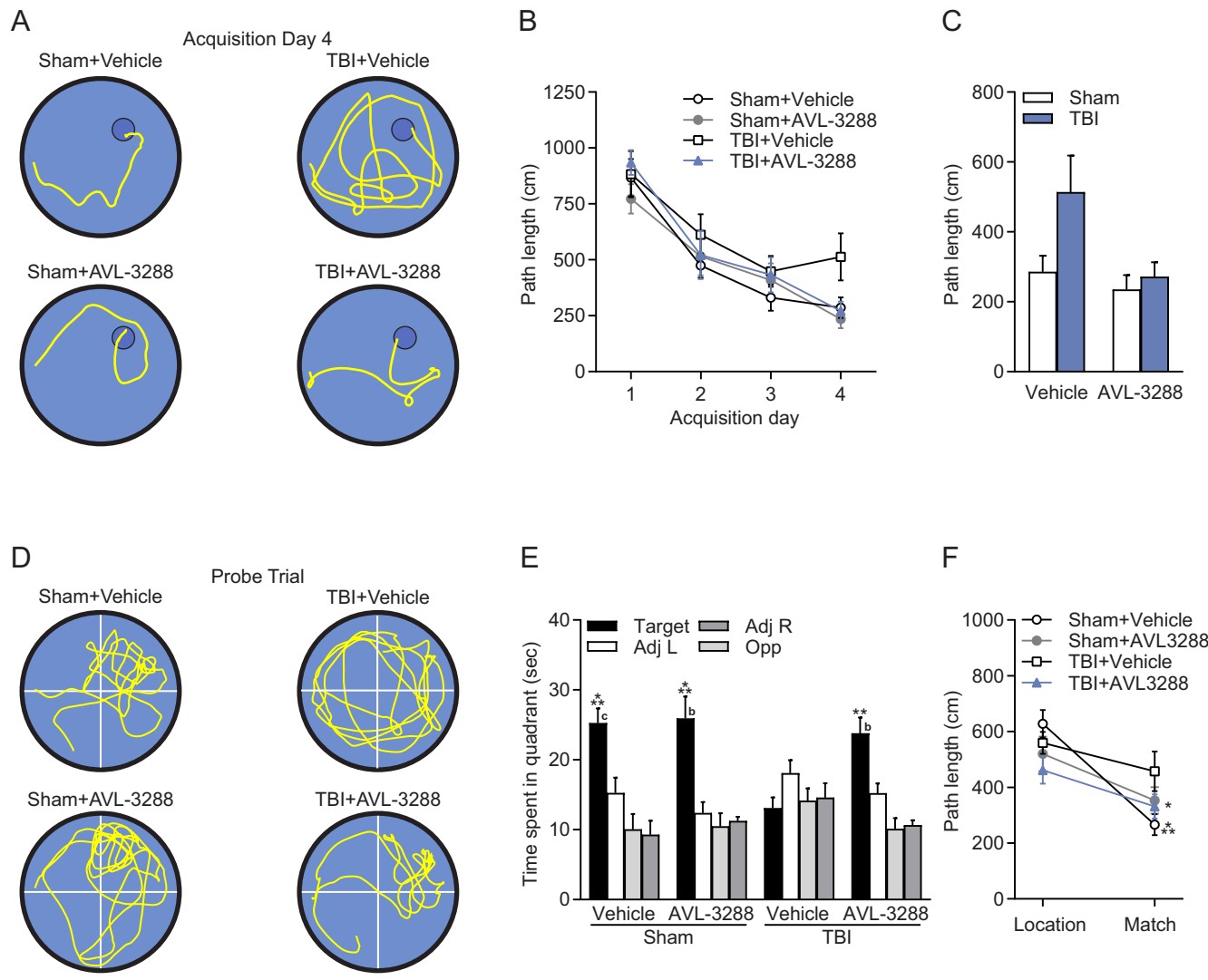

**Fig 3. Water maze.** AVL-3288 treatment improved spatial memory in TBI animals. A) Representative swim tracks from acquisition day 4. B) Swim path length during water maze acquisition. C) Path length on acquisition day 4. D) Representative swim tracks during the probe trial with the platform removed from the northeast, target quadrant. E) Time spent in each quadrant of the water maze during the probe trial. $^{**}p<0.01$, $^{***}p<0.001$ vs. Other Quadrants, $^{b}p<0.01$, $^{c}p<0.001$ vs. TBI+Vehicle Target Quadrant, two-way ANOVA with Tukey's HSD post-hoc test. F) Path length to reach the platform on the location and match trials, with a 5 s interval between trials to assess working memory. $^{*}p<0.05$ for Sham+AVL-3288 and TBI+AVL-3288 Location vs. Match trials, $^{***}p<0.001$ for Sham+Vehicle Location vs. Match trials, repeated measures two-way ANOVA with Tukey's HSD post-hoc test. Mean ± SEM, $n = 10$ Sham+Vehicle, $n = 8$ Sham +AVL-3288, $n = 11$ TBI+Vehicle, $n = 11$ TBI+AVL-3288.

treatment. At 3 months post-surgery, the time point corresponding to the beginning of behavioral testing, acute hippocampal slices were generated and recordings were made in stratum radiatum of area CA1 (Fig 5). Hippocampal slices were treated with vehicle or AVL-3288 (1 μM). Afferent fiber excitability as measured by the fiber volley amplitude was unaltered after TBI, but synaptic strength as assessed with the fEPSP slope was significantly depressed (animal group x current intensity interaction: $F_{(30,195)} = 2.79$, $p<0.001$). Analysis of the relationship between the fEPSP slope and fiber volley amplitude indicated a significant decrease of the fEPSP response in relation to the fiber volley amplitude in TBI+Vehicle slices, and this was rescued with AVL-3288 treatment (surgery x drug treatment interaction: $F_{(1,20)} = 4.45$,

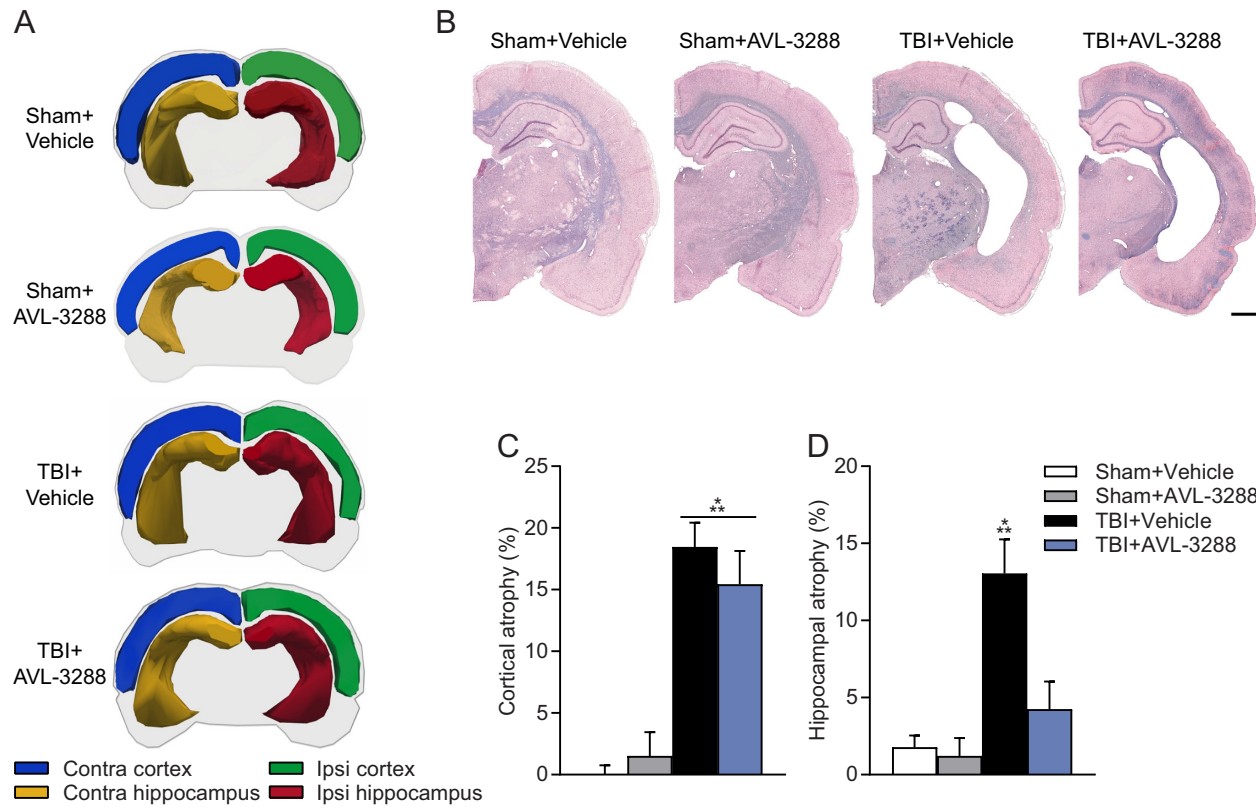

**Fig 4. Pathology.** Atrophy levels of the cortex and hippocampus. A) Representative 3-dimensional visualizations of brains reconstructed from contours of the ipsilateral (Ipsi) and contralateral (Contra) cortex and hippocampus between bregma levels -3.3 to -6.8 mm. B) Representative H&E plus Luxol fast blue-stained sections at bregma level -3.0 mm. Cortical atrophy was similar between TBI+Vehicle and TBI+AVL-3288 animals. Hippocampal atrophy was modestly reduced in TBI+AVL-3288 animals. *Scale bar* 1 mm. C) Quantification of atrophy of the ipsilateral parietal cortex. ***$p$<0.001 vs. Sham, two-way ANOVA with Tukey's HSD post-hoc test. D) Quantification of atrophy of the ipsilateral hippocampus. ***$p$<0.001 vs. Sham+Vehicle, Sham+AVL-3288, TBI+AVL-3288, two-way ANOVA with Tukey's HSD post-hoc test. Mean ± SEM, $n$ = 10 Sham +Vehicle, $n$ = 8 Sham+AVL-3288, $n$ = 11 TBI+Vehicle, $n$ = 11 TBI+AVL-3288.

$p$ = 0.0478). PPF was not significantly different between slices from sham animals or TBI animals treated with vehicle or AVL-3288 (S1 Fig). AVL-3288 treatment prior to and during high frequency stimulation rescued the deficits in LTP maintenance. The initial phase of LTP, 1–5 min after high frequency stimulation (HFS), was similar across treatment groups. The maintenance phase of LTP, 30–60 min after HFS, was significantly reduced in slices from TBI animals, and this was improved with AVL-3288 treatment (surgery x drug treatment interaction: $F_{(1,23)}$ = 4.39, $p$ = 0.047). No significant effects were observed with the level of depolarization during HFS between treatment groups (S1 Fig). Additionally, synaptic fatigue during the HFS was not significantly different between treatment groups (S1 Fig).

## Discussion

In this study, we tested whether a positive allosteric modulator of the α7 nAChR, AVL-3288, would reduce chronic cognitive deficits in a preclinical model of TBI. Using the moderate FPI model in rats, we found that AVL-3288 treatment prior to training on several learning tasks reduced memory deficits in cue and contextual fear conditioning, as well as spatial memory retention in the water maze and spatial working memory. The effects of AVL-3288 were long-lasting, with a short treatment of AVL-3288 during training on cue and fear conditioning

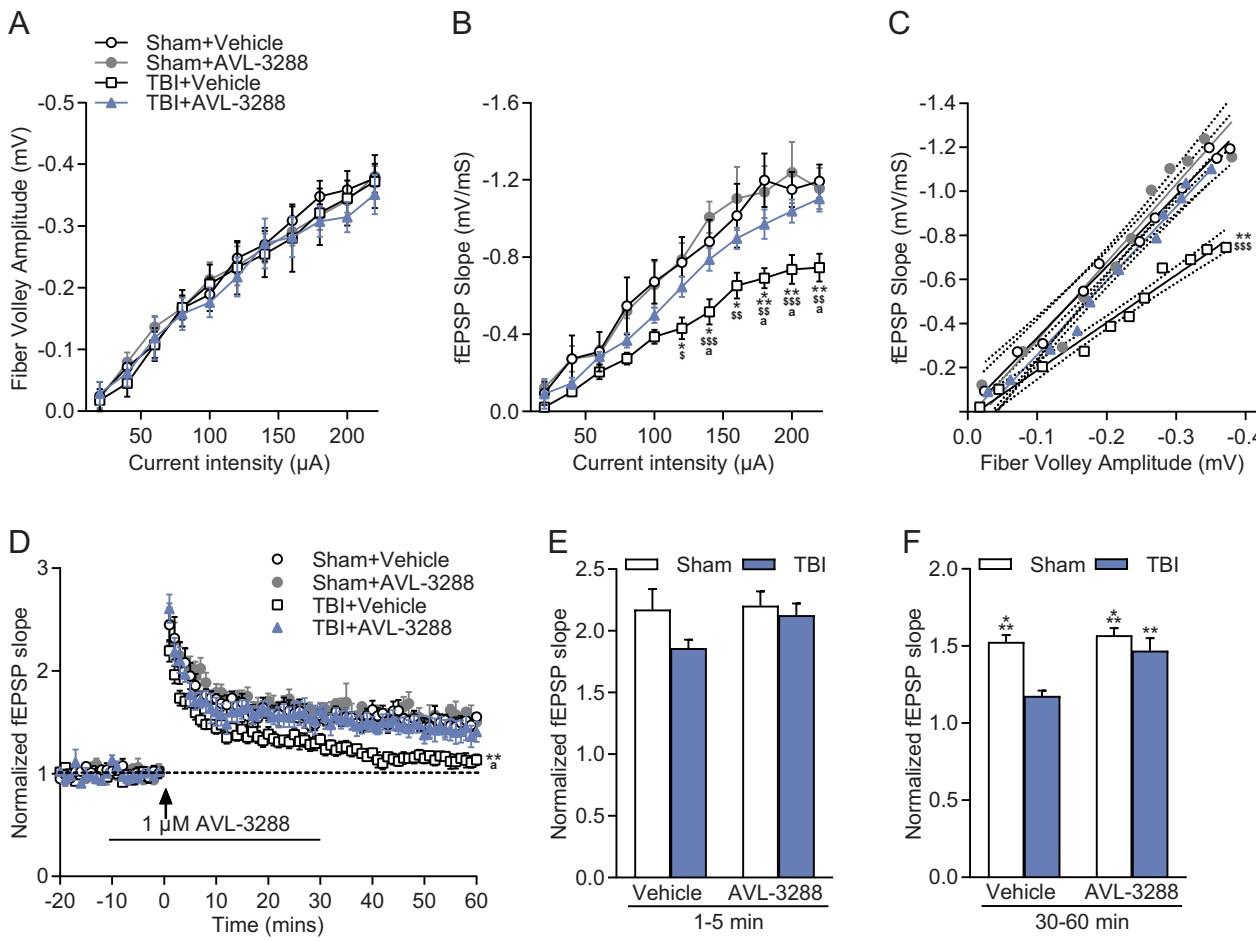

**Fig 5. Electrophysiology.** Improvement of basal synaptic transmission and rescue of hippocampal LTP with AVL-3288 treatment. A) I-O curves in stratum radiatum of area CA1 of the fiber volley amplitude and stimulus intensity. B) I-O curves of the fEPSP slope and stimulus intensity. $^{*}p < 0.05$, $^{**}p < 0.01$, $^{***}p < 0.001$ vs. Sham+Vehicle, $^{a}p < 0.05$ vs. TBI+AVL-3288, $^{\$}p < 0.05$, $^{\$\$}p < 0.01$, $^{\$\$\$}p < 0.001$ vs. Sham+AVL-3288, repeated measures two-way ANOVA with Tukey's HSD post-hoc test. C) I-O curves of the fiber volley amplitude and fEPSP slope relationship. *Dotted lines* depict 95% confidence intervals. $^{**}p < 0.01$ vs. Sham+Vehicle, TBI+AVL-3288, $^{\$\$\$}p < 0.001$ vs. Sham+AVL-3228, two-way ANOVA of linear regression fit slopes. D) Hippocampal LTP in area CA1. $^{**}p < 0.01$ vs. Sham+Vehicle, Sham+AVL-3288, $^{a}p < 0.05$ vs. TBI+AVL-3288, repeated measures two-way ANOVA with Tukey's HSD post-hoc test. E) Effects of AVL-3288 on the initial phase of LTP, 1–5 min after HFS. F) Effects of AVL-3288 on the maintenance phase of LTP, 30–60 min after HFS. $^{**}p < 0.01$, $^{***}p < 0.001$ vs. TBI+Vehicle, two-way ANOVA with Tukey's HSD post-hoc test. Mean ± SEM, $n = 5$ slices/5 Sham +Vehicle animals, $n = 5$ slices/5 Sham+AVL-3288 animals, $n = 7$ slices/6 TBI+Vehicle animals, $n = 7$–8 slices/7 TBI+AVL-3288 animals.

resulting in a long-lasting improvement in retention of fear memory at 1 month post-training. AVL-3288 treatment of hippocampal slices from TBI animals also improved LTP and synaptic transmission in area CA1. Hippocampal atrophy, but not cortical atrophy after TBI was reduced with AVL-3288. These results demonstrate that AVL-3288 is a promising therapeutic to treat cognitive deficits after TBI.

AVL-3288 is a type 1 positive allosteric modulator that has recently been evaluated in a phase 1 clinical trial for safety and pharmacokinetics in healthy subjects [51]. AVL-3288 has an EC$_{50}$ of 0.7 μM for human α7 nAChR in *Xenopus* oocytes, a brain plasma ratio of 2-fold and half-life of 7.5 hr with oral dosing in mice [47]. The cardiovascular and respiratory safety profile in humans has been evaluated and no adverse effects have been observed at doses up to 30 mg which results in plasma levels over 4-fold greater than that observed in the current study [51]. AVL-3288 is selective for α7 nAChRs, having minimal activity towards other nAChRs, or

the homologous $GABA_A$ α1β2γ2L receptor and the 5-$HT_{3A}$ receptor [47]. In this study, we found that single administration of AVL-3288 at 0.3 mg/kg intraperitoneally reached brain levels of 3 μM within 30 min of administration. This dose is 4-fold higher than the $EC_{50}$ for enhancing human α7 nAChR currents in *Xenopus* oocytes [47]. These results suggest that the drug may have achieved levels high enough to modulate α7 nAChR conductance, although an important limitation of this study is that we did not directly measure α7 nAChR currents. Another important limitation of this study is that AVL-3288 levels were measured in naïve, non-injured animals, but not in TBI animals which may alter the brain distribution and levels of AVL-3288. Although the elimination rate in plasma in humans is known to be 3 hr, the rate of clearance within the rat brain has not yet been determined [51]. Future studies using PET ligands would be highly useful to measure brain distribution, target engagement, and rate of clearance [51].

Since systemically administered AVL-3288 reached levels in the brain that could potentially alter α7 nAChR currents by 30 min after treatment, we chose to treat animals 30 min prior to training on each learning task. Each behavior task took 1–2 hr to complete daily, therefore this drug administration protocol was chosen to optimize availability of the drug within the brain at the time of learning. Administering AVL-3288 to be on board at the time of learning, but not during retention, resulted in a persistent memory enhancing effect, improving both water maze retention in the probe trial at 24 hr post-training and cue and contextual fear conditioning at both 24 hr and 1 month post-training. These results are in accordance with a previous study demonstrating that AVL-3288 treatment in naïve rats enhanced social preference of a juvenile, novel rat when assessed 24 hr after drug administration [56]. Further studies are aimed at determining the duration of the effects of AVL-3288 treatment on memory enhancement after TBI.

However, in contrast to other previous studies, we did not observe any nootropic effects of AVL-3288 on sham animals during acquisition or retention of either the water maze task or cue and contextual fear conditioning [47, 57, 58]. In these prior studies, AVL-3288 administration to naïve rats improved performance in the radial arm maze at a similar dose to what was used in this study [47]. Repeated dosing with a higher concentration of AVL-3288 in naïve rats also enhanced novel social preference [56]. Our differing results may be due to the spaced treatment protocol since the sham and TBI animals only received 8 doses of AVL-3288 distributed over a 1 month time period. Alternatively, the memory paradigms that we chose may not have been sensitive enough to detect a memory enhancing effect in sham animals, or acetylcholine levels were sufficient for maximal memory encoding in sham animals with these particular tasks.

Beyond hippocampal effects, we also observed a significant improvement in cue fear conditioning. This result suggested that positive allosteric modulation of α7 nAChRs in TBI animals may have resulted in a generalized improvement in learning and memory ability. Alternatively, direct effects within the amygdala may explain this result. Less is known about how α7 nAChRs are affected in the amygdala as compared to the hippocampus. There is evidence in a mild TBI model that surface expression and currents of α7 nAChRs are actually upregulated in the amygdala after mild controlled cortical impact injury in rats [59]. These changes were associated with a loss of GABAergic interneurons and increased excitability in the basolateral amygdala. In the FPI model or controlled cortical impact, acetylcholine or α7 nAChR levels are unaltered in the amygdala, suggesting that changes in acetylcholine or α7 nAChRs after TBI within the amygdala may be model-dependent [60–62]. Further experiments are needed to determine if AVL-3288 altered α7 nAChR currents within the amygdala with this TBI model.

Consistent with our behavioral results and previous studies evaluating positive allosteric modulators of α7 nAChRs in naïve rats, AVL-3288 did not significantly alter basal synaptic transmission in area CA1 in sham, non-injured animals [20, 63]. In TBI animals, we found that AVL-3288 rescued the depression in synaptic strength as indicated by an improvement in the fEPSP–fiber volley relationship. In addition, we found that the decaying LTP in slices from TBI animals was rescued with AVL-3288 treatment. Further studies are needed to determine the duration of the enhancement in LTP. Positive allosteric modulators and partial agonists of α7 nAChRs facilitate LTP induction and convert short-term potentiation into non-decaying LTP [20, 21, 64]. This facilitation of LTP induction is due to both pre- and postsynaptic mechanisms, by increasing glutamate release as characterized by an increase in miniature excitatory postsynaptic current frequency, as well as increased postsynaptic depolarization leading to the activation of extracellular signal-regulated kinase 1/2 (ERK1/2) and cAMP response element-binding protein (CREB) [64–66]. We observed no differences in PPF or the fiber volley amplitude with TBI or AVL-3288 treatment, suggesting that the effects of AVL-3288 were post-synaptic. These results are in partial accordance with a previous study using the controlled cortical impact model that reported a recovery of the fiber volley amplitude by 7 days post-injury [67]. Together, these results suggest that AVL-3288 may have improved learning and memory ability in TBI animals by facilitating postsynaptic signaling, leading to the activation of ERK1/2 and CREB, which are well known to be dysregulated and contribute to memory deficits after TBI [52, 68, 69].

An intriguing finding of our study is the reduction in hippocampal atrophy with AVL-3288 treatment of TBI animals. There are several potential mechanisms of why AVL-3288 had an effect on hippocampal atrophy, but did not significantly alter cortical atrophy. There are regional differences in the distribution of α7 nAChRs in the brain. α7 nAChRs are highly expressed in the hypothalamus, moderately in hippocampus and striatum, and at low levels in the midbrain, cortex and cerebellum [70, 71]. In the hippocampus, the α7 nAChR subtype is the most highly expressed nAChR subtype, and the effects of nicotine are predominantly through α7 nAChRs in the CA1 region [12, 16, 72, 73]. These studies suggest that AVL-3288 may have had a greater effect in the hippocampus than the cortex due to the distribution of nAChR subtypes. In accordance with our results, a previous study found that intermittent nicotine administration rescued hippocampal-dependent learning impairments after controlled cortical impact, but had no significant effects on reducing cortical lesion volume [34]. It is well established that TBI results in chronic inflammation [74]. α7 nAChRs are expressed on microglia and α7 nAChR agonists have known anti-inflammatory effects both within the brain and the periphery [75–79]. We have observed a persistent upregulation of microglia within the brain after 3–6 months post-injury and this increase was greater in the hippocampus than the cortex [68]. Further experiments are needed to address the differential effects of AVL-3288 in the hippocampus and cortex and determine if AVL-3288 may have reduced inflammation in the hippocampus after TBI.

Several clinical trials for TBI have been implemented with limited success using the cholinesterase inhibitors rivastigmine, donepezil and galantamine [24, 40–42, 44, 80–83]. In Alzheimer's disease, cholinesterase inhibitors have also been tested in combination with memantine with a few studies suggesting beneficial effects were greater with galantamine than with donepezil [84–87]. Donepezil is a cholinesterase inhibitor whereas galantamine is both a cholinesterase inhibitor as well as a positive allosteric modulator of α4β2 and α7 nAChRs. These studies support the possibility that the combinatorial treatment of AVL-3288 with memantine may be a promising therapeutic approach for TBI [84, 88]. This approach has support at the circuitry level given the cooperation of cholinergic and glutamatergic signaling to convert short term plasticity into long-lasting LTP [89, 90]. Another approach that merits

consideration is the FDA-approved smoking cessation drug varenicline. Varenicline is a high-affinity partial agonist of the α4β2 nAChR and a full agonist of α7 nAChRs [91]. However, varenicline did not improve cognition in either patients with schizophrenia or mild-to-moderate Alzheimer's disease [92, 93]. This study and other studies solely targeting the α7 nAChR for schizophrenia and Alzheimer's disease further support the importance of considering combinatorial treatments [94]. Given the inflammatory processes still evident in the chronic recovery phase of TBI, the combination of an anti-inflammatory treatment such as minocycline and AVL-3288 may provide synergistic effects [68, 95, 96]. Minocycline in combination with N-acetylcysteine (NAC) has shown efficacy in improving cognition in schizophrenia patients [97].

AVL-3288 as a positive allosteric modulator of the α7 nAChR has a significant advantage over clinically used cholinesterase inhibitors. Cholinesterase inhibitors result in homeostatic changes within the brain due to increased levels of agonist by upregulation and/or downregulation of receptor levels [54, 98–103]. An important advantage of AVL-3288 is that repeated administration did not upregulate α7 nAChR levels, even when given once daily at 10 mg/kg for 7 days [56]. Additionally, AVL-3288 has no effect on α7 nAChR currents unless an endogenous agonist is present, preserving the endogenous spatial and temporal kinetics of cholinergic modulation of learning and memory circuits [47]. As a type I positive allosteric modulator, AVL-3288 does not affect the rapid desensitization kinetics of the α7 nAChR unlike type II modulators [47]. Type II modulators block or slow desensitization and thus, have the potential to be cytotoxic and disrupt synaptic/extra-synaptic plasticity mechanisms regulated by use-dependent receptor desensitization [104, 105]. These advantageous properties of AVL-3288 and our findings indicate that positive allosteric modulation of α7 nAChRs with AVL-3288 is a promising therapeutic strategy to facilitate cognitive recovery in chronic phase of TBI.

## Supporting information

**S1 Fig. PPF, depolarization levels during HFS, and synaptic fatigue during HFS were not altered after TBI or AVL-3288 treatment.** A) PPF ratio in response to stimulus intervals from 12.5–250 msec. B) Total depolarization levels during HFS and steady-state depolarization levels during the last 50 ms of HFS. C) Synaptic fatigue during HFS. Mean ± SEM, $n = 5$–6 slices/5-6 Sham+Vehicle animals, $n = 5$-6/5-6 Sham+AVL-3288 animals, $n = 7$ slices/6 TBI+Vehicle animals, $n = 7$–8 slices/7 TBI+AVL-3288 animals.
(EPS)

## Acknowledgments

We thank Dr. Concepcion Furones and Dr. Oscar Alcazar for assistance with animal surgeries, Ms. Chantal Sanchez for critical reading of the manuscript, and Yan Shi and The Miami Project to Cure Paralysis microscopy core for technical support.

## Author Contributions

**Conceptualization:** David J. Titus, Timothy Johnstone, Coleen M. Atkins.

**Data curation:** David J. Titus, Timothy Johnstone, Nathan H. Johnson, Sidney H. London, Meghana Chapalamadugu, Coleen M. Atkins.

**Formal analysis:** David J. Titus, Nathan H. Johnson, Sidney H. London, Meghana Chapalamadugu, Coleen M. Atkins.

**Funding acquisition:** Coleen M. Atkins.

**Investigation:** David J. Titus, Timothy Johnstone, Nathan H. Johnson, Coleen M. Atkins.

**Methodology:** David J. Titus, Timothy Johnstone, Nathan H. Johnson, Sidney H. London, Meghana Chapalamadugu, Derk Hogenkamp, Kelvin W. Gee, Coleen M. Atkins.

**Project administration:** Coleen M. Atkins.

**Resources:** Derk Hogenkamp, Kelvin W. Gee, Coleen M. Atkins.

**Software:** Coleen M. Atkins.

**Supervision:** David J. Titus, Kelvin W. Gee, Coleen M. Atkins.

**Validation:** Coleen M. Atkins.

**Visualization:** David J. Titus, Nathan H. Johnson, Sidney H. London, Meghana Chapalamadugu, Coleen M. Atkins.

**Writing – original draft:** David J. Titus, Timothy Johnstone, Nathan H. Johnson, Sidney H. London, Meghana Chapalamadugu, Derk Hogenkamp, Kelvin W. Gee, Coleen M. Atkins.

**Writing – review & editing:** David J. Titus, Timothy Johnstone, Nathan H. Johnson, Sidney H. London, Meghana Chapalamadugu, Derk Hogenkamp, Kelvin W. Gee, Coleen M. Atkins.

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
