## [Decision Letter · Decision Letter 0]

1 Jul 2019

PONE-D-19-14787

Positive allosteric modulation of the α7 nicotinic acetylcholine receptor as a treatment for cognitive deficits after traumatic brain injury

PLOS ONE

Dear Dr. Atkins,

Thank you for submitting your manuscript to PLOS ONE. After careful consideration, we feel that it has merit but does not fully meet PLOS ONE’s publication criteria as it currently stands. Therefore, we invite you to submit a revised version of the manuscript that addresses the points raised during the review process.

We would appreciate receiving your revised manuscript by Aug 15 2019 11:59PM. To enhance the reproducibility of your results, we recommend that if applicable you deposit your laboratory protocols in protocols.io, where a protocol can be assigned its own identifier (DOI) such that it can be cited independently in the future. For instructions see: http://journals.plos.org/plosone/s/submission-guidelines#loc-laboratory-protocols

We look forward to receiving your revised manuscript.

Kind regards,

Alexandra Kavushansky, PhD

Academic Editor

PLOS ONE

**Journal Requirements:**

"I have read the journal's policy and the authors of this manuscript have the following competing interests: K.W.G., T.B.J. and D.J.H. are inventors on patents covering the compound AVL-3288 and uses thereof. "

3.

We note that you have included the phrase “data not shown” in your manuscript. Unfortunately, this does not meet our data sharing requirements. PLOS does not permit references to inaccessible data. We require that authors provide all relevant data within the paper, Supporting Information files, or in an acceptable, public repository. Please add a citation to support this phrase or upload the data that corresponds with these findings to a stable repository (such as Figshare or Dryad) and provide and URLs, DOIs, or accession numbers that may be used to access these data. Or, if the data are not a core part of the research being presented in your study, we ask that you remove the phrase that refers to these data.

**Comments to the Author**

1. Is the manuscript technically sound, and do the data support the conclusions?

Reviewer #1: Yes

Reviewer #2: Partly

Reviewer #3: Partly

2. Has the statistical analysis been performed appropriately and rigorously? 

Reviewer #1: I Don't Know

Reviewer #2: Yes

Reviewer #3: Yes

3. Have the authors made all data underlying the findings in their manuscript fully available?

Reviewer #1: Yes

Reviewer #2: Yes

Reviewer #3: Yes

4. Is the manuscript presented in an intelligible fashion and written in standard English?

Reviewer #1: Yes

Reviewer #2: Yes

Reviewer #3: Yes

5. Review Comments to the Author

Reviewer #1: This is an interesting study looking at positive allosteric modulation of the α7 nicotinic acetylcholine receptor as a treatment for cognitive impairments after TBI. This has clinical relevance and is a major unmet need.

I am not a basic scientist – hence, cannot comment on animal experiments methodology and other details.

Consider adding the following to enrich the discussion.

https://www.ncbi.nlm.nih.gov/pubmed/29570959

I don’t see any papers on varenicline and traumatic brain injury – but merits a discussion considering the mechanism of action of varenicline.

Another potential treatment for TBI. One medication is unlikely to be effective.

https://www.ncbi.nlm.nih.gov/pubmed/?term=bergold+nac+minocycline

https://www.ncbi.nlm.nih.gov/pubmed/30776665 This paper discusses the limitation of the minocycline-NAC combination and a potential solution.

https://www.ncbi.nlm.nih.gov/pmc/articles/PMC4173077/figure/F1/

https://link.springer.com/article/10.1007/s40473-019-00174-5

https://www.ncbi.nlm.nih.gov/pubmed/?term=panacea+bailey

https://www.ncbi.nlm.nih.gov/pubmed/343145 neuropsychological tests couldn’t distinguish cognitive dysfunction in schizophrenia vs. after TBI.

https://www.ncbi.nlm.nih.gov/pubmed/28250730

https://www.ncbi.nlm.nih.gov/pubmed/21254300

https://www.ncbi.nlm.nih.gov/pubmed/?term=massey+2006+bdnf

https://www.ncbi.nlm.nih.gov/pubmed/28065843 hence, targeting glutamate/NMDA mechanism concurrently may be needed to detect a clinically meaningful signal.

We found that AVL-3288

95 improved cognitive deficits when administered 3 months after experimental TBI and

96 rescued hippocampal LTP deficits. This line doesn’t belong where it is.

Future studies may be done with PET ligands. https://www.ncbi.nlm.nih.gov/pubmed/29522184

Also, measure target engagement biomarkers. See Figure 1. https://link.springer.com/article/10.1007/s40473-019-00174-5

Another caveat of this study is that AVL-3288 levels were

405 measured in naïve animals, but not in TBI animals. Not clear to me – animals didn’t have TBI?

galantamine in the CREATE trial (NCT01416948). Is this not published yet? No citation available?

Find out whether galantamine studies were better than donepezil and rivastigmine. Even if it was not significantly better, that would be an important point to highlight that the benefit was from the nicotinic action.

Is the sample size adequate? If so, mention it.

Reviewer #2: This study investigates the effects of AV-3288 on cognitive functions after TBI in rats. AV-3288 is a type I positive allosteric modulator (PAM) of α7 nAChRs and may act as a therapeutic agent after TBI by augmenting the brain cholinergic α7-dependent signaling. TBI reduces expression of functional α7 nAChRs but does not eliminate it. Thus, treatments that augment cholinergic transmission (e.g., PAMs) may become beneficial after TBI. The results indicate that AV-3288 administered 3 months after fluid-percussion brain injury significantly improves performance of male rats in Morris Water Maze (MWM) and cue/contextual fear conditioning (FC) assays. Histological assays indicated that AV-3288 reduces hippocampal but not cortical atrophy. Finally, in electrophysiological experiments in acute hippocampal slices obtained from rats 3 months after TBI, TBI reduced and AV-3288 rescued the magnitude of LTP during the maintenance phase. The study concluded that AV-3288 improves cognitive function and hippocampal integrity and function after TBI in male rats. The results are informative despite a few issues that need to be addressed.

Major issues:

Each animal was subjected to multiple rounds of isoflurane anesthesia which is neuroprotective. Was the total cumulative duration of anesthesia monitored to keep it equal across groups? Fig. 2B suggests that the same animals were used for FC and MWM assays. A clear statement should be provided which should also include the number of AV-3288 injections and anesthesia rounds that each animal received over the course of experiments.

What was the rationale for combining sham+vehicle and sham+AV-3288 groups? I do not believe it is appropriate to pool treated and untreated groups under any circumstances. Insignificant differences between these groups do not make them identical and may affect comparisons with TBI groups.

There is no discussion of differences in the effects of AV3288 in hippocampus vs. cortex (Fig.4). Fig.4A needs more explanation as all panels look very similar. In Fig.4B, examples of TBI+vehicle and TBI+AV-3288 -treated slices look very similar and do not seem to represent differences in hippocampal atrophy summarized in Fig. 4D.

Stimulus intensities alone do not define synaptic responses. Both axonal excitability (Fiber Volley, FV) and synaptic strength may be affected by TBI and/or treatments. A relationship between fEPSP slope and FV in sham vs. TBI groups would be more informative. Does TBI alter the FV magnitudes?

Other specific issues:

Figure 1 is not very informative as it does not allow evaluating the rate of clearance.

How were the physiological parameters (shown in Table 1) measured?

If vehicle contained 2% DMSO and 8% Solutol, what was the other 90%? Was it a typo and should have been 20% and 80%?

The conclusions do not seem to do justice to the key finding that a short treatment with AV-3288 3 months after TBI significantly enhances cognitive function in rats and this effect is long-lasting as evidenced by repeated measurements 1 month later. I recommend making a clear summary statement like that upfront and in the abstract.

This may be a far reach, but does LTP magnitude remain elevated in AV3288-treated animals after TBI 1 month after treatment?

Reviewer #3: The study by Titus et al. investigates effects of behavioral and synaptic transmission recovery of function after experimental brain trauma by administering rats multiple injections of a positive allosteric modulator at the alpha7 nicotinic acetylcholine receptors. AVL-3288. Results are interesting, but the authors are invited to address a series of questions meant to enhance the quality of the manuscript and data reporting.

There are only 4-5 published papers with AVL-3288 in rodents and they use primarily 1 mg/kg for behavioral benefits. Whilte the authors report brain levels to be possibly sufficient to modulate alpha7 nAChR conductance, they also acknowledge that the experiment was performed in naive animals. How can they reason that the choice of 0.3 mg/kg may be clinically relevant ?

AVL was administered prior to all behavioral tasks except memory retention. Why was this task left out and tested drug free?

Injections were commenced at nearly 3 months post surgery, which is long past secondary injury occurring. How do the authors reason that histology is improved with just a few injections that long after injury, when rats were sacrificed after the last injection? This treatment would seem quite effective compared to many others administered to rodents chronically and in the acute phase post TBI. Moreover, the 7-8 injections are not delivered rhythmically enough (aka daily for example) in order to render steady state levels of drug in the bloodstream (which would be clinically relevant).

Combining Sham groups may create an overpowered analysis with 18 rats in the Sham group. Can the authors stand by their results if analyses are run with all 4 groups, with both Injury and Injection as factors? (Two way ANOVA)

6. PLOS authors have the option to publish the peer review history of their article (what does this mean?). If published, this will include your full peer review and any attached files.

Reviewer #1: Yes: Maju Koola

Reviewer #2: No

Reviewer #3: No

---

## [Author Response · Author response to Decision Letter 0]

23 Aug 2019

August 14, 2019

Manuscript number: PONE-D-19-14787

Title: Positive allosteric modulation of the α7 nicotinic acetylcholine receptor as a treatment for cognitive deficits after traumatic brain injury

Alexandra Kavushansky, PhD

Academic Editor

PLOS ONE

Dear Dr. Kavushansky,

Thank you very much for the review of our manuscript. We greatly appreciate the reviewers’ positive comments and their suggested changes. Reviewer #1 appreciated the significance of this research and stated, “This has clinical relevance and is a major unmet need.” Reviewers 2 and 3 found the results “informative” and “interesting.” The reviewers have provided highly constructive suggestions for improvement and we have revised the manuscript accordingly to address all of the reviewers’ comments. Below are our responses to the reviewers’ comments and detailed descriptions of the revisions. The revisions within the manuscript are indicated with tracked changes as requested by the Journal.

Journal Requirements

Comment 1. Ensure that your manuscript meets PLOS ONE’s style requirements for filing naming, text formatting and authors’ affiliations.

Response: We have ensured that the resubmitted manuscript and file names are in compliance with PLOS ONE’s style requirements. 

Comment 2. Confirm that your Competing Interests section adheres to PLOS ONE policies on sharing data and materials. 

Response: We have revised our Competing Interests statement as suggested. The statement is now, “I have read the journal’s policy and the authors of this manuscript have the following competing interests: K.W.G., T.B.J. and D.J.H. are inventors on patents covering the compound AVL-3288 and uses thereof. This does not alter our adherence to PLOS ONE policies on sharing data and materials.”

Comment 3. The phrase “data not shown” does not meet PLOS ONE data sharing requirements. 

Response: This phrase has been removed from this section since it is no longer relevant. The two treatment groups have now been analyzed separately as suggested by Reviewers 2 and 3. We had also used the phrase “not shown” regarding data in reference to Fig 5. This data is now provided as a supplementary figure (S1 Fig). 

Reviewer 1

Comment 1. Consider adding the following to enrich the discussion. https://www.ncbi.nlm.nih.gov/pubmed/29570959

Response: Thank you for the suggestion and we have now added this citation to the discussion which now includes a discussion of the potential of combinatorial treatments of AVL-3288 with other promising therapeutics such as memantine. 

Comment 2. I don’t see any papers on varenicline and traumatic brain injury – but merits a discussion considering the mechanism of action of varenicline.

Response: To our knowledge, there are no published preclinical or clinical studies of varenicline and traumatic brain injury. We appreciate this suggestion given the mechanism of action, and have now included this potential therapeutic in the discussion. 

Comment 3. Another potential treatment for TBI. One medication is unlikely to be effective.

https://www.ncbi.nlm.nih.gov/pubmed/?term=bergold+nac+minocycline
https://www.ncbi.nlm.nih.gov/pubmed/30776665 This paper discusses the limitation of the minocycline-NAC combination and a potential solution.

https://www.ncbi.nlm.nih.gov/pmc/articles/PMC4173077/figure/F1/

https://link.springer.com/article/10.1007/s40473-019-00174-5

https://www.ncbi.nlm.nih.gov/pubmed/?term=panacea+bailey
https://www.ncbi.nlm.nih.gov/pubmed/343145 neuropsychological tests couldn’t distinguish cognitive dysfunction in schizophrenia vs. after TBI.

https://www.ncbi.nlm.nih.gov/pubmed/28250730

https://www.ncbi.nlm.nih.gov/pubmed/21254300

https://www.ncbi.nlm.nih.gov/pubmed/?term=massey+2006+bdnf

https://www.ncbi.nlm.nih.gov/pubmed/28065843 hence, targeting glutamate/NMDA mechanism concurrently may be needed to detect a clinically meaningful signal.

Response: We thank the reviewer for bringing to our attention these other potential combinatorial therapeutic strategies. The discussion has been expanded to include a paragraph about the need to consider combinatorial treatments and in particular, targeting the glutamatergic system as suggested by several of these papers. 

Comment 4. Lines 95 and 96 do not belong here in the introduction.

Response: We have removed this conclusion from the introduction and an overview of the findings of the paper is included at the start of the discussion. 

Comment 5. Future studies may be done with PET ligands. Also, measure target engagement biomarkers.

Response: We appreciate this suggestion and have now included this in the discussion. 

Comment 6. Another caveat of this study is that AVL-3288 levels were measured in naïve animals, but not in TBI animals. Not clear to me – animals didn’t have TBI?

Response: We apologize for the lack of clarity. Yes, the animals that were used to measure AVL-3288 levels did not have a TBI. We now more clearly state that AVL-3288 was measured in naïve, non-injured animals. We appreciate that this is a limitation of our study and this is more clearly stated in the discussion. 

Comment 7. Is the CREATE trial using galantamine published yet (NCT01416948)?

Response: To our knowledge, the results from the CREATE trial are not published. We have removed the clinical trial reference and now provide an earlier citation of a clinical trial testing galantamine for TBI. 

Comment 8. Find out whether galantamine studies were better than donepezil and rivastigmine. Even if it was not significantly better, that would be an important point to highlight that the benefit was from the nicotinic action.

Response: There are only a few studies that have directly compared these combinatorial treatments, and the results have been mixed. These clinical trials are now included in the discussion. 

Comment 9. Is the sample size adequate? If so, mention it.

Response: Yes, the sample size was adequate and we now mention this in the methods. We state in the methods section that we performed an a priori power analysis and determined that a sample size of 11 animals/group would detect a 20% difference in water maze probe trial performance between groups at 80% power. The effect size for our observed differences in the water maze probe trial was 1.28, recent recall of contextual fear conditioning was 1.39, remote recall of contextual fear conditioning was 0.92 and LTP expression at 30-60 post-tetanization was 1.27. These large effect sizes indicate that our sample size was adequate for the main objective of this study, to determine if AVL-3288 improved hippocampal learning and LTP in TBI animals. 

Reviewer 2

Comment 1. Each animal was subjected to multiple rounds of isoflurane anesthesia which is neuroprotective. Was the total cumulative duration of anesthesia monitored to keep it equal across groups? 

Response: Yes, the cumulative duration of anesthesia was similar for both sham and TBI animals, as well as between drug treatment groups. We now provide a statement in the methods that indicates that we monitored the anesthesia duration to keep the anesthesia duration consistent across all treatment groups. 

Comment 2. Fig. 2B suggests that the same animals were used for FC and MWM assays. A clear statement should be provided which should also include the number of AVL-3288 injections and anesthesia rounds that each animal received over the course of experiments.

Response: Yes, the animals were serially tested first for fear conditioning and then for water maze performance. This is more clearly stated in the figure legend. The number of AVL-3288 injections and anesthesia rounds are now clearly stated in the methods section. Each animal received two anesthesia rounds for the TBI surgery, which was 3 months prior to behavioral testing. The number of injections each animal received are depicted in Fig 2A by the arrows and stated in the figure legend. 

Comment 3. What was the rationale for combining sham+vehicle and sham+AVL-3288 groups? I do not believe it is appropriate to pool treated and untreated groups under any circumstances. Insignificant differences between these groups do not make them identical and may affect comparisons with TBI groups.

Response: We appreciate the reviewer’s criticism, and this was also suggested by Reviewer 3 in comment 4. We have now reanalyzed and re-graphed the data separating the Sham+Vehicle and Sham+AVL-3288 groups for Figs 2-5. The statistical analyses have now been revised using a two-way ANOVA with the factors surgery and drug treatment. A significant interaction of surgery x drug treatment was observed for recent and remote recall of cue and contextual fear conditioning, water maze probe trial retention, fEPSP slope vs. fiber volley amplitude, LTP expression, and hippocampal atrophy. These significant interactions of surgery and drug treatment are now reported in the results section. 

Comment 4. There is no discussion of differences in the effects of AV3288 in hippocampus vs. cortex (Fig. 4). Fig. 4A needs more explanation as all panels look very similar. In Fig. 4B, examples of TBI+vehicle and TBI+AVL-3288-treated slices look very similar and do not seem to represent differences in hippocampal atrophy summarized in Fig. 4D.

Response: We have provided more description of the images in the results section and figure legend for Fig 4. The images look very similar likely because cortical atrophy was not significantly different in TBI+Vehicle and TBI+AVL-3288 treated animals. Furthermore, hippocampal atrophy was only modestly reduced in TBI+AVL-3288 treated animals (mean differences 8.8%). These results are more clearly stated in the results section and figure legend. 

We have now revised the discussion to include several potential mechanisms of why AVL-3288 had an effect on hippocampal atrophy, but did not significantly alter cortical atrophy. There are regional differences in the distribution of α7 nicotinic acetylcholine receptors (nAChRs) in the brain. α7 nAChRs are highly expressed in the hypothalamus, moderately in hippocampus and striatum, and at low levels in the midbrain, cortex and cerebellum (Maier et al., Neuropharm 2011; Nirogi et al., J Pharm Toxicol Meth 2012). In the hippocampus, the α7 nAChR subtype is the most highly expressed nAChR subtype, and the effects of nicotine are predominantly through α7 nAChRs in the CA1 region (Han et al., Eur J Neuro 2000; Breese et al., J Comp Neurol 1997; Seguela et al., J Neuro 1993; Orr‐Urtreger et al., J Neuro 1997). These studies suggest that AVL-3288 may have had a greater effect in the hippocampus than the cortex due to the distribution of nAChR subtypes. In accordance with our results, a previous study found that intermittent nicotine administration rescued hippocampal-dependent learning impairments after controlled cortical impact, but had no significant effects on reducing cortical lesion volume (Verbois et al., Neurosci 2003). Further experiments are needed to address the differential effects of AVL-3288 in the hippocampus and cortex.

Comment 5. Stimulus intensities alone do not define synaptic responses. Both axonal excitability (Fiber Volley, FV) and synaptic strength may be affected by TBI and/or treatments. A relationship between fEPSP slope and FV in sham vs. TBI groups would be more informative. Does TBI alter the FV magnitudes?

Response: We appreciate this comment and have now reanalyzed and replotted the input-output responses to evaluate the relationship of fEPSP slope and fiber volley. The fiber volley amplitude was not altered, but the fEPSP slope was significantly decreased with TBI. Analysis of the relationship between the fEPSP slope and fiber volley amplitude indicated a significant decrease of the fEPSP response in relation to the fiber volley amplitude in TBI+Vehicle slices, and this was rescued with AVL-3288 treatment. These results have now been added to Fig 5 and we now discuss these results in the discussion. These results are in partial accordance with a previous study that observed a reduction in the fiber volley amplitude at 2 days post-injury, which recovered by 7 days post-injury (Norris et al., J Neurotrauma 2009). This citation has now been added to the discussion. 

Comment 6. Figure 1 is not very informative as it does not allow evaluating the rate of clearance.

Response: We appreciate the reviewer’s comment and state this limitation in the discussion. The elimination rate of AVL-3288 from plasma has been published in humans and this is now included in the discussion. However, the rate of clearance from the rat brain has not yet been determined. 

Comment 7. How were the physiological parameters (shown in Table 1) measured?

Response: We now provide more methodology in the methods section regarding the physiological measurements. All animals received catheter implantation in the tail artery. Mean arterial blood pressure (MAPB) was monitored continuously during the surgery with a transducer connected to the tail artery catheter and LabChart 7 software (ADInstruments). Blood gases and blood pH were measured from blood harvested from the tail artery with a blood gas analyzer (ABL800 Flex, Radiometer America). Atmospheres of pressure were measured using a transducer connected to the fluid percussion injury shaft and LabChart 7 software.

Comment 8. If vehicle contained 2% DMSO and 8% Solutol, what was the other 90%? Was it a typo and should have been 20% and 80%.

Response: We apologize for the lack of clarity and have provided more description regarding the final composition of the vehicle in the methods section. The vehicle contained 2% DMSO, 8% solutol and 90% saline.

Comment 9. The conclusions do not seem to do justice to the key finding that a short treatment with AV-3288 3 months after TBI significantly enhances cognitive function in rats and this effect is long-lasting as evidenced by repeated measurements 1 month later. I recommend making a clear summary statement like that upfront and in the abstract.

Response: We appreciate the reviewer’s comment and now state this finding more clearly in the abstract and first paragraph of the discussion. 

Comment 10. This may be a far reach, but does LTP magnitude remain elevated in AV3288-treated animals after TBI 1 month after treatment?

Response: This is an excellent point of the reviewer and unfortunately we did not extend our analyses of LTP to one month after treatment. Determining the duration of the effects of AVL-3288 treatment is important as we consider developing this therapy further and is a focus of future study. This point is now more emphasized in the discussion.

Reviewer 3

Comment 1. There are only 4-5 published papers with AVL-3288 in rodents and they use primarily 1 mg/kg for behavioral benefits. While the authors report brain levels to be possibly sufficient to modulate alpha7 nAChR conductance, they also acknowledge that the experiment was performed in naive animals. How can they reason that the choice of 0.3 mg/kg may be clinically relevant?

Response: We now provide more rationale for the dose selection in the results section. The minimum effective dose to improve radial arm maze memory in rats was 0.3 mg/kg (Ng et al., PNAS 2007). In a Phase 1 clinical study, levels of all doses tested in humans reached levels obtained with 0.3 mg/kg rats with the radial arm maze (see Fig 1 in Gee et al., J Psychopharm 2017). 

Comment 2. AVL was administered prior to all behavioral tasks except memory retention. Why this task was left out and tested drug free?

Response: AVL-3288 treatment was omitted for memory retention for both fear conditioning and water maze to determine if the drug needed to be on board for the enhancement of learning and memory. This rationale was guided by previous in vitro studies that demonstrated that activation of α7 nAChRs converted hippocampal short-term potentiation into LTP (Nakauchi et al., Euro J Neuro 2012 #4780; Lagostena et al., Neuropharm, 2008). This rationale is now more clearly described in the results section. 

Comment 3. Injections were commenced at nearly 3 months post-surgery, which is long past secondary injury occurring. How do the authors reason that histology is improved with just a few injections that long after injury, when rats were sacrificed after the last injection? This treatment would seem quite effective compared to many others administered to rodents chronically and in the acute phase post TBI. Moreover, the 7-8 injections are not delivered rhythmically enough (aka daily for example) in order to render steady-state levels of drug in the bloodstream (which would be clinically relevant).

Response: We were also surprised to observe a small reduction in hippocampal atrophy with this delayed and intermittent treatment in the chronic recovery period. We have now expanded the discussion to discuss this finding and potential mechanisms. It is well established that TBI results in chronic inflammation (Ramlackhansingh et al., Annal Neurol 2011). We have observed a persistent upregulation of microglia within the brain after 3-6 months post-injury and this increase was greater in the hippocampus than the cortex (Titus et al., J Neuro 2016). α7 nAChR agonists have anti-inflammatory effects within the brain and the periphery after TBI. Further experiments are needed to determine if AVL-3288 may have reduced inflammation in the hippocampus after TBI. 

Comment 4. Combining Sham groups may create an overpowered analysis with 18 rats in the Sham group. Can the authors stand by their results if analyses are run with all 4 groups, with both Injury and Injection as factors? (Two way ANOVA).

Response: We appreciate the reviewer’s criticism, and this was also suggested by Reviewer 2 in comment 3. The analyses have now been run with all 4 groups, with a two-way ANOVA and the factors surgery and drug treatment. A significant interaction of surgery and drug treatment was obtained for recent and remote recall of cue and contextual fear conditioning, water maze probe trial retention, fEPSP slope vs. fiber volley amplitude, LTP expression, and hippocampal atrophy. These significant interactions of surgery and drug treatment indicate that AVL-3288 treatment improved hippocampal learning and memory and LTP in TBI animals. The significant interactions are now reported in the results section and the separated data are now presented in revised Figs 2-5. 

We thank the reviewers for their suggested revisions which have improved the manuscript. We hope these revisions have made the paper suitable for publication in PLOS ONE.

Sincerely,

Coleen M. Atkins, Ph.D.

Associate Professor

Department of Neurological Surgery

The Miami Project to Cure Paralysis

University of Miami Miller School of Medicine

---

## [Decision Letter · Decision Letter 1]

17 Sep 2019

Positive allosteric modulation of the α7 nicotinic acetylcholine receptor as a treatment for cognitive deficits after traumatic brain injury

PONE-D-19-14787R1

Dear Dr. Atkins,

We are pleased to inform you that your manuscript has been judged scientifically suitable for publication and will be formally accepted for publication once it complies with all outstanding technical requirements.

With kind regards,

Alexandra Kavushansky, PhD

Academic Editor

PLOS ONE

Additional Editor Comments (optional):

Reviewers' comments:

Reviewer's Responses to Questions

**Comments to the Author**

1. If the authors have adequately addressed your comments raised in a previous round of review and you feel that this manuscript is now acceptable for publication, you may indicate that here to bypass the “Comments to the Author” section, enter your conflict of interest statement in the “Confidential to Editor” section, and submit your "Accept" recommendation.

Reviewer #2: All comments have been addressed

2. Is the manuscript technically sound, and do the data support the conclusions?

Reviewer #2: Yes

3. Has the statistical analysis been performed appropriately and rigorously? 

Reviewer #2: Yes

4. Have the authors made all data underlying the findings in their manuscript fully available?

Reviewer #2: Yes

5. Is the manuscript presented in an intelligible fashion and written in standard English?

Reviewer #2: Yes

6. Review Comments to the Author

Reviewer #2: The authors adequately addressed all of my comments..................................................

7. PLOS authors have the option to publish the peer review history of their article (what does this mean?). If published, this will include your full peer review and any attached files.

Reviewer #2: Yes: Victor V. Uteshev

---

## [Editor Report · Acceptance letter]

25 Sep 2019

PONE-D-19-14787R1 

Positive allosteric modulation of the α7 nicotinic acetylcholine receptor as a treatment for cognitive deficits after traumatic brain injury 

Dear Dr. Atkins:

I am pleased to inform you that your manuscript has been deemed suitable for publication in PLOS ONE. Congratulations! Your manuscript is now with our production department. 

With kind regards,

on behalf of

Dr. Alexandra Kavushansky 

Academic Editor

PLOS ONE